# Amelanotic Melanocytic Nevus of the Oral Cavity: A Case Report and Literature Review

**DOI:** 10.3390/diagnostics15121554

**Published:** 2025-06-18

**Authors:** Rossana Izzetti, Filippo Minuti, Angela Pucci, Chiara Cinquini, Antonio Barone, Marco Nisi

**Affiliations:** 1Department of Surgical Pathology, Medicine, Molecular and Critical Area, University of Pisa, Via Roma 67, 56126 Pisa, Italy; rossana.izzetti@unipi.it (R.I.); f.minuti3@studenti.unipi.it (F.M.); angelapucci@libero.it (A.P.); chiara.cinquini@gmail.com (C.C.); antonio.barone@unipi.it (A.B.); 2Dental Biomaterials Research Unit (d-BRU), University of Liege, B-4020 Liege, Belgium

**Keywords:** nevus, gingiva, diagnosis, oral, oral medicine, oral surgical procedures

## Abstract

Amelanotic melanocytic nevi of the oral cavity are uncommon lesions that often present a diagnostic challenge for clinicians, primarily due to their nonspecific clinical appearance and the broad spectrum of possible differential diagnoses. These lesions can mimic a variety of benign and malignant conditions, requiring precise histopathological confirmation. The primary objective of this article is to present a comprehensive case report—tracing the course from initial presentation through diagnostic workup to final diagnosis—and to provide an overview of the current literature on oral amelanotic melanocytic nevi. We report the case of a 27-year-old female who presented with a small, exophytic mass located in the anterior mandibular gingival region. The lesion was asymptomatic and lacked pigmentation, adding to the diagnostic uncertainty. A range of differential diagnoses was considered, including pyogenic granuloma, peripheral ossifying fibroma, and squamous cell carcinoma. Due to the lesion’s limited size and accessibility, an excisional biopsy was performed under local anesthesia. Histopathological examination revealed an amelanotic melanocytic nevus, a rare variant characterized by the absence of melanin pigment, further complicating the clinical impression. The diagnosis was confirmed through immunohistochemical staining, which demonstrated melanocytic markers consistent with a nevus. The patient was followed up with for six months postoperatively, with no evidence of recurrence or malignant transformation. This case highlights the critical role of biopsy in achieving a definitive diagnosis, especially in lesions with atypical clinical presentations. It also underscores the importance of considering amelanotic melanocytic nevi in the differential diagnosis of nonpigmented oral lesions, as well as maintaining vigilance regarding the rare possibility of amelanotic melanoma.

## 1. Introduction

Oral melanocytic nevi are relatively uncommon lesions with unknown origin, with a reported prevalence ranging from 0.1% to 0.5% in the general population [1,2], representing the most frequent form of melanocytic proliferation within the oral mucosa [3]. Among the various histopathological subtypes, the intramucosal nevus is the most prevalent, accounting for approximately 55% to 80.6% of all cases [4,5,6]. This subtype is characterized by nests or cords of nevus cells located within the lamina propria, beneath an intact epithelium, often without significant clinical pigmentation. Oral intramucosal nevi have a very low annual incidence, estimated at 4.35 cases per 10 million persons, underscoring their rarity in clinical practice [5]. This low frequency contributes to the potential for misdiagnosis, especially when lesions present without the typical pigmentary features associated with melanocytic proliferations [1].

The second most common subtypes are the compound nevus and the blue nevus. Pigmentation is present in 80–87% of cases, depending on the sample examined, while the remaining cases present as amelanotic nevi. Other variants may also appear amelanotic, although at negligible frequencies.

Amelanotic lesions pose significant diagnostic challenges due to their lack of pigmentation, often mimicking both benign and malignant conditions across various anatomical sites in the oral cavity [7]. Differential diagnoses include fibroma, pyogenic granuloma, papilloma, mucocele, and early melanoma, while cutaneous presentations require distinction from dermatofibroma, basal cell carcinoma, and amelanotic melanoma [8]. Beyond these common locations, amelanotic nevi have been documented in several sites. Ocular manifestations include the choroidal amelanotic nevus, appearing as a pale fundus lesion that must be differentiated from choroidal melanoma, and the conjunctival amelanotic nevus, which may resemble pinguecula or early squamous neoplasia [9,10]. In the nasal cavity and sinuses, these lesions can be mistaken for inflammatory polyps or sinonasal melanoma. Genital mucosal sites may present with amelanotic lesions that mimic lichen planus or early melanoma [11]. Rare cases have also been reported in the esophagus and anorectal mucosa, where they must be distinguished from gastrointestinal stromal tumors or metastatic deposits [12]. The diagnostic complexity is compounded by clinical variability, as some initially amelanotic lesions may develop pigmentation over time, and their rarity in certain locations increases the risk of misdiagnosis. Advanced diagnostic tools including dermoscopy, reflectance confocal microscopy, ultrasonography, optical coherence tomography (for ocular lesions), and immunohistochemical markers (S100, HMB-45, SOX10) are crucial for accurate identification, particularly in high-risk locations where melanoma must be excluded [13,14,15,16,17].

Amelanotic melanocytic nevi are rare, benign proliferations of melanocytes that lack visible pigmentation, making them diagnostically challenging. Amelanotic nevi represent a smaller subset, with approximately 20% of oral melanocytic nevi lacking pigmentation. The prevalence of amelanotic nevi varies depending on the histological subtype, with 22–32% of intramucosal nevi being amelanotic [5]. These lesions are more commonly observed in females, with a peak incidence in the third decade of life. Amelanotic melanocytic nevi typically present as asymptomatic, well-demarcated, exophytic, or flat lesions, usually measuring less than 1 cm in diameter [18]. The most common sites of occurrence include the hard palate, vestibular mucosa, vermilion border of the lips, and gingiva [6]. Due to the absence of pigmentation, these lesions can be easily mistaken for other benign oral lesions, such as fibromas, papillomas, or reactive gingival lesions [2]. The lack of pigmentation often leads to delayed diagnosis, as clinicians may not initially consider a melanocytic origin in the differential diagnosis [19].

Amelanotic nevi may appear as smooth or slightly raised lesions with a pink or red hue, resembling other common oral lesions such as squamous papilloma or pyogenic granuloma [20]. The absence of pain or bleeding further complicates the clinical diagnosis, as these lesions are often discovered incidentally during routine dental examinations. In some cases, focal pigmentation may be present, which can aid in the diagnosis, but the overall clinical appearance remains nonspecific [21]. In terms of lesion morphology, approximately two-thirds of melanocytic nevi appear raised, while the remaining third are flat. Nearly 80% of intramucosal nevi are elevated, whereas blue nevi and compound nevi are typically flat [4,5]. There are limited data on the clinical morphology of amelanotic nevi. Ferreira et al. [4] found that 45% of amelanotic nevi were raised, while 55% were flat. When nevi are both pigmented and elevated, differentiation from other pigmented oral lesions, such as melanotic macules or exogenous/endogenous pigmentations, is facilitated since these typically present as flat lesions [22].

Histologically, amelanotic melanocytic nevi are characterized by the well-differentiated proliferation of melanocytes within the connective tissue, typically devoid of mitotic activity [23]. Immunohistochemical staining is often employed to confirm the diagnosis, with markers such as MART-1 and p16 being positive, while HMB45 and Ki67 are usually negative [24]. These findings help to differentiate amelanotic nevi from other melanocytic lesions, including melanoma [24]. The absence of mitotic activity and the lack of cellular atypia are key features that distinguish amelanotic nevi from more aggressive lesions such as melanoma [23]. However, the possibility of amelanotic melanoma should always be considered, particularly in cases with rapid growth or atypical clinical features [8].

The diagnosis of amelanotic melanocytic nevi is particularly challenging, given their rarity and the overlap in clinical presentation with other benign or malignant oral lesions. As such, definitive diagnosis often relies on histopathological examination and immunohistochemical analysis. The primary objective of this article is to present a comprehensive case report, detailing the progression from the patient’s initial clinical presentation to the final histopathological diagnosis, and to review and synthesize the current evidence regarding oral amelanotic melanocytic nevi.

## 2. Case Report

A 27-year-old female was referred to the Unit of Dentistry and Oral Surgery at the University Hospital of Pisa for the evaluation of an exophytic mass located in the anterior mandibular gingival region. Her medical history was unremarkable, with no known systemic conditions, no ongoing pharmacological therapies, and no history of tobacco use. The patient reported that the lesion had been present for approximately three months, exhibiting a progressive increase in size without episodes of regression, exacerbation, pain, or bleeding. She also denied any history of trauma to the affected area.

Intraoral examination revealed a sessile, exophytic, oval-shaped lesion measuring approximately 4 mm × 8 mm, located on the vestibular attached gingiva adjacent to teeth 3.1 and 3.2 (Figure 1). The lesion extended up to, but did not cross, the mucogingival junction, and there was no involvement of the gingival margin or interdental papilla between the two teeth. The surface appeared multilobulated, with no evident color variation compared to the surrounding mucosa, although faint brownish pigmentation was observed in the adjacent gingival tissue. On palpation, the lesion was soft and nontender. Thermal pulp testing confirmed the vitality of teeth 3.1 and 3.2. Periodontal probing yielded no pathological findings, and periapical radiographs showed no significant abnormalities.

Given the lesion’s limited dimensions, an excisional biopsy was performed (Figure 2).

Histopathological examination revealed a well-differentiated subepithelial melanocytic proliferation devoid of mitotic activity. Immunohistochemical analysis demonstrated MART-1 and p16 positivity, with no immunoreactivity for HMB45 or Ki67, corroborating the benign nature of the melanocytic lesion (Figure 3).

The patient was followed postoperatively to ensure uneventful healing. A six-month follow-up assessment revealed the complete restoration of the gingival architecture with no evidence of recurrence (Figure 4). No further intervention was deemed necessary.

## 3. Discussion

Considering the lesion’s clinical characteristics and the absence of osseous involvement, the following differential diagnoses were contemplated: verruciform xanthoma, peripheral ameloblastoma, linear epidermal nevus, squamous papilloma, and amelanotic melanocytic nevus.

### 3.1. Verruciform Xanthoma

Verruciform xanthoma (VX) is an uncommon lesion predominantly affecting the oral mucosa, with a predilection for males in the fifth to sixth decades of life. Its etiopathogenesis remains uncertain, although chronic inflammation, trauma, and immune dysregulation have been postulated as contributing factors [25,26]. The gingiva, palate, and alveolar mucosa are frequently involved. Clinically, VX manifests as a raised plaque or nodular lesion with a papillary or verrucous surface, well-demarcated margins, and a color range spanning from white to red. The clinical differentiation of VX is challenging due to its resemblance to epithelial lesions such as squamous cell papilloma. Histopathologic analysis is essential, typically demonstrating lipid-laden foamy macrophages (“xanthoma cells”) within epithelial rete ridges and connective tissue papillae [27,28]. The standard treatment modality is surgical excision, with recurrence being exceedingly rare.

### 3.2. Peripheral Ameloblastoma

Peripheral ameloblastoma (PA) is a rare odontogenic neoplasm, constituting 1–10% of all ameloblastomas [29]. Unlike its intraosseous counterparts, PA is confined to the gingival soft tissue. It is hypothesized to originate from remnants of the dental lamina. PA predominantly affects the mandibular gingiva and presents as a sessile, exophytic lesion with a smooth or verrucous surface, pink to red in hue, and asymptomatic. Histologically, it exhibits epithelial cell organization in follicular or cord-like patterns, akin to conventional ameloblastomas [30]. Due to its nonspecific clinical attributes, PA is often misdiagnosed as pyogenic granuloma, peripheral giant cell granuloma, ossifying fibroma, or squamous papilloma. Given the lesion’s morphology and anatomical distribution, PA was considered a plausible differential diagnosis despite its rarity.

### 3.3. Linear Epidermal Nevus

Linear epidermal nevus (LEN) is an ectodermal hamartoma, primarily affecting the skin, with oral manifestations being exceedingly rare [31]. LEN typically arises at birth, expands during childhood, and stabilizes in adolescence. It can appear as a unilateral linear lesion or as discrete papules and plaques, clinically mimicking HPV-associated entities such as squamous papilloma and verruca vulgaris. However, LEN retains the coloration of the surrounding mucosa and exhibits distinct growth kinetics. Histopathologically, it is characterized by papillomatous projections with hyperkeratosis and acanthosis, sharply delineated from a normal epithelium [32]. While the clinical features of LEN aligned with those of the current lesion, its typical early onset and rare oral involvement rendered this diagnosis improbable.

### 3.4. Squamous Papilloma

Squamous papilloma (SP) is among the most prevalent benign oral epithelial proliferations, often associated with HPV infection [33]. It is most frequently encountered between the third and seventh decades of life. Commonly affected sites include the palate and tongue, although lesions may arise anywhere within the oral cavity. SP typically presents as an exophytic lesion with a verrucous, “cauliflower-like” surface, ranging from white to pink/red, often pedunculated and generally less than 5 mm in diameter. Due to its clinical similarity to HPV-associated lesions such as verruca vulgaris and condyloma acuminatum, histopathological evaluation is crucial for accurate diagnosis [34]. Although the lesion’s attributes did not perfectly align with those of HPV-related lesions, SP was included among the differential diagnoses due to its high incidence.

### 3.5. Amelanotic Melanocytic Nevus

These lesions are more commonly observed in females, with peak incidence around the third decade of life. The hard palate, vestibular mucosa, vermilion border of the lips, and gingiva are the most frequently affected sites. Clinically, they present as either flat or raised lesions, typically measuring less than 1 cm in diameter [4]. Approximately 20% of oral melanocytic nevi lack pigmentation, complicating differentiation from fibromas and papillomas [6]. In the current case, focal pigmentation within the lesion suggested the possibility of an amelanotic melanocytic nevus.

In the present case, the final diagnosis of an oral amelanotic melanocytic nevus was confirmed by the histopathological examination.

The diagnostic process of oral melanocytic nevi becomes significantly complex when dealing with amelanotic variants, which lack the characteristic pigmentation that typically raises clinical suspicion [5]. These nonpigmented lesions can mimic common benign oral lesions such as fibromas or traumatic ulcers, thus necessitating a thorough and systematic approach to diagnosis that incorporates both clinical and histopathological evaluation [2,4].

The diagnostic uncertainty surrounding amelanotic nevi underscores the critical importance of histopathological evaluation. All suspicious oral lesions, particularly those demonstrating rapid growth (>3 mm/month), irregular borders, or unusual clinical features (e.g., surface ulceration, bleeding), should undergo biopsy without exception [3,6]. This principle becomes especially relevant when considering the possibility of oral melanoma, a rare but aggressive malignancy that accounts for approximately 0.5% of all oral cancers and carries a 5-year survival rate of only 15–20% [25,28]. While the majority (80–87%) of oral pigmented lesions are benign melanotic macules or nevi, no clinical parameters exist to date that can reliably differentiate early melanoma from its benign counterparts [7,9]. Certain anatomical locations, particularly the hard palate (accounting for 40–50% of cases) and maxillary gingiva, should always raise heightened clinical concern due to their well-documented association with melanoma development [19,35].

The current understanding of oral melanoma pathogenesis suggests that 70–75% of cases arise de novo, without any identifiable precursor lesion [35,36]. However, approximately 25–30% of cases may develop from pre-existing pigmented lesions that have been present for extended periods (median duration 5.2 years before malignant transformation [2,37]. Despite this observation, the scientific community has not reached a consensus regarding the potential of oral melanocytic nevi to undergo malignant transformation. Large cohort studies analyzing over 100 cases each have found no statistically significant association between oral nevi and subsequent melanoma development [6,22]. The available evidence does not currently support the notion that oral nevi represent a significant risk factor for melanoma development, and their precise role in oral carcinogenesis remains poorly understood [6,23].

The diagnostic challenge becomes even more pronounced when considering amelanotic melanoma, an exceptionally rare variant representing about 2.3–5% of all oral melanomas [38,39]. These lesions present a particular diagnostic dilemma due to their lack of visible pigmentation, often leading clinicians to initially consider more common benign conditions such as pyogenic granuloma or peripheral giant cell granuloma [8,24]. The conventional designation of “amelanotic” can be somewhat misleading, as careful histopathological examination frequently reveals at least focal melanin deposition in 60–70% of cases [25,40]. In such cases, immunohistochemical analysis using markers such as HMB-45 (positive in 85–90% of melanomas), S-100 (95–100% sensitivity), and MART-1 (75–80% specificity) becomes indispensable for accurate diagnosis and appropriate management [23,26]. Recent evidence has also highlighted the potential diagnostic value of the SOX10 and MITF markers in challenging cases [40].

The management of oral melanocytic nevi, particularly amelanotic variants, requires a balanced approach that considers both diagnostic certainty and therapeutic necessity. Complete excisional biopsy with 2–3 mm clinical margins serves as the diagnostic gold standard, providing both therapeutic removal and definitive histopathological evaluation [28]. For confirmed benign lesions, no additional treatment is typically required following complete excision, with recurrence rates reported at <5% in long-term follow-up studies [39]. The management of positive surgical margins remains a topic of ongoing discussion, with current practice suggesting that re-excision may be warranted in cases demonstrating severe dysplasia (defined as >3 atypical melanocytes per high-power field), while lesions with mild to moderate dysplasia may be appropriately managed through clinical observation given their low recurrence potential (2–7% at 5 years) [41].

Long-term clinical follow-up (every 6–12 months for 3–5 years) assumes particular importance for lesions exhibiting atypical features or those with incomplete initial excision [14]. Current management protocols are largely derived from dermatologic practice, where the management of dysplastic nevi continues to evolve, although oral mucosal lesions may demonstrate different biological behavior [42]. The prevailing consensus suggests a conservative approach for mild to moderately dysplastic lesions through clinical monitoring (visual inspection and photographic documentation), reserving re-excision for cases demonstrating severe dysplasia, mitotic activity (>1/mm^2^), or other concerning features, such as ulceration or rapid growth [41]. Indeed, prospective studies have shown recurrence rates of only 3–5% for benign to moderately dysplastic nevi followed for 5 years, suggesting that aggressive management may not be necessary in all cases [14,42].

Amelanotic melanocytic nevi of the oral cavity present unique diagnostic challenges and potential for clinical confusion with both benign and malignant conditions [5,18]. The absence of characteristic pigmentation often leads to their exclusion from initial differential diagnoses, with studies reporting an average diagnostic delay of 6–8 months compared to pigmented lesions [4,8]. Histopathological examination, supplemented by immunohistochemical analysis in ambiguous cases, remains essential for definitive diagnosis and appropriate management [23]. While these lesions are typically benign, clinical vigilance is warranted, particularly for lesions located at high-risk sites such as the palate (40% of oral melanomas) or those demonstrating rapid growth (>2 mm/month) [36]. The current lack of standardized management protocols highlights the need for further research to establish evidence-based guidelines for these uncommon oral lesions.

## 4. Conclusions

Amelanotic melanocytic nevi are rare, benign oral lesions that require careful clinical and histopathological evaluation to ensure accurate diagnosis and appropriate management. The lack of pigmentation and nonspecific clinical features can lead to diagnostic challenges, highlighting the importance of biopsy and immunohistochemical analysis. While these lesions are generally benign, clinicians should remain vigilant regarding the rare possibility of amelanotic melanoma, particularly in cases with rapid onset or atypical features. Further research is needed to refine the diagnostic criteria and management protocols for amelanotic melanocytic nevi in the oral cavity.

## Figures and Tables

**Figure 1 diagnostics-15-01554-f001:**
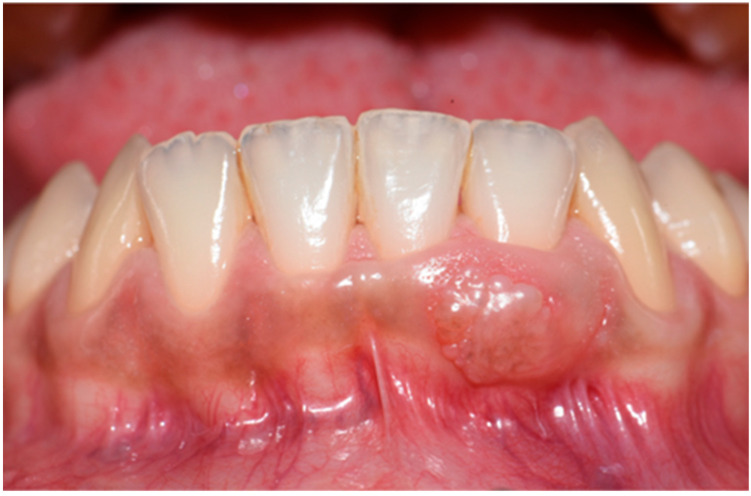
Clinical appearance of the gingival lesion. An exophytic mass of the attached gingiva can be observed at the level of teeth 3.1–3.2. The lesion presents with an oval shape and lobulated surface.

**Figure 2 diagnostics-15-01554-f002:**
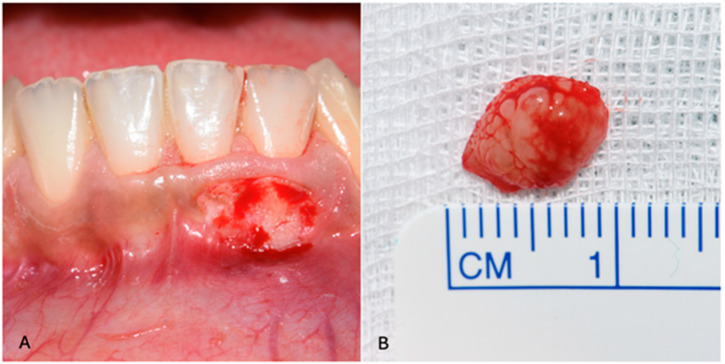
(**A**) Excisional biopsy of the lesion and (**B**) removed fragment of approximately 8 mm × 4 mm.

**Figure 3 diagnostics-15-01554-f003:**
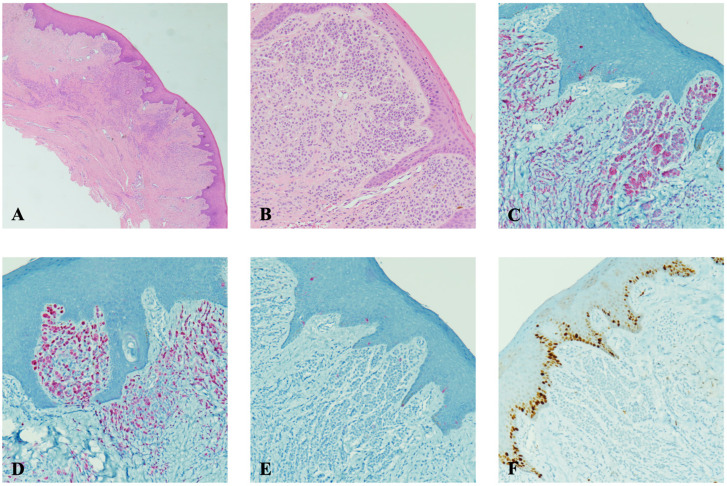
The amelanotic melanocytic nevus of the oral mucosa at low (**A**) and high (**B**) magnification. No significant deposition of melanin is detected by histology, showing the well-differentiated sub-epithelial melanocytic proliferation with no mitotic figure. By immunohistochemistry, the nevus shows MART1 (**C**) and p16 (**D**) positivity with no immunoreactivity for HMB45 (**E**) or proliferating index Ki67 (**F**), with such an immunostaining pattern confirming the benign nature of the melanocytic neoformation. (**A**,**B**): Hematoxylin and eosin histochemical staining; original magnification: 2× (**A**) and 10× (**B**). (**C**–**F**): Immunoperoxidase stainings with hematoxylin counterstaining; original magnification: 10×.

**Figure 4 diagnostics-15-01554-f004:**
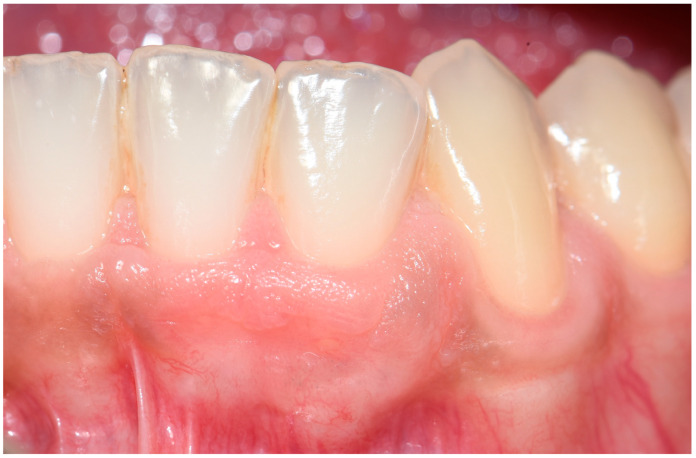
Clinical situation at follow-up. No recurrence of the pathology was detected, with complete resolution in the absence of additional treatment.

## Data Availability

All available data are contained within the article.

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
