# Peer review of "Amelanotic Melanocytic Nevus of the Oral Cavity: A Case Report and Literature Review"

_diagnostics, 2025, doi:10.3390/diagnostics15121554_

Round 1

Reviewer 1 Report

Comments and Suggestions for Authors

The article aims to discuss a case of amelanotic nevus and a literature review on differential diagnoses. There are some points that deserve to be reviewed by the authors:

  1. The abstract and introduction deal with melanotic nevus. I think it would be best to mention only amelanotic nevus, which is the focus of the article, in order to meet the article's objectives.
  2. The introduction is too long.
  3. The clinical part of the case report is separated from the anatomopathological description, with a review between the two topics. I suggest doing the clinical and microscopic description together and a topic for the literature review with differential diagnoses.
  4. The discussion is repetitive of the introduction and contains more information about melanocytic nevus than amelanotic nevus. I suggest reworking the discussion, focusing on differential diagnosis, rarity, the diagnostic process and treatment.
  5. There are only 2 specific references to amelanotic nevus. I understand that it is a rare lesion, but there needs to be more specific references to this lesion and not to melanotic nevus.

Author Response

Please find our Notes below.

Reviewer 2 Report

Comments and Suggestions for Authors

Dear Authors,

thank you for the interesting article. Here are some suggestions:

  1. In the introduction, describe other melanotic changes of oral cavity, that should be brought to the Readers' attention (differencial diagnosis) - not only fibromas, papillomas, or reactive gingival lesions
  2. Note that the origin is not known
  3. In the discussion, add the aspect of other mucosal changes, especially within diseases and health conditions - think that many changes are red in colour and this one is different, which is an interesting fact. Also, refer to miscolourations due to metals present in the oral cavity (eg. amalgam), but also metal poisoning. Also refer to medicine taken and use of rinses.
  4. Also, when you describe facts in the discussion - refer them to the case, eg. you describe that most of the patients have de novo changes. What about your Patient?

Thank you

Author Response

Please find our notes below.

Round 2

Reviewer 1 Report

Comments and Suggestions for Authors

The authors made the suggested corrections, and the article improved significantly, being accepted for publication.